# The Age of Information in Wireless Cellular Systems: Gaps, Open Problems, and Research Challenges

**DOI:** 10.3390/s23198238

**Published:** 2023-10-03

**Authors:** Elena Zhbankova, Abdukodir Khakimov, Ekaterina Markova, Yuliya Gaidamaka

**Affiliations:** 1Department of Probability Theory and Cyber Security, Peoples’ Friendship University of Russia (RUDN University), 6 Miklukho-Maklaya Str., Moscow 117198, Russia; 2Federal Research Center “Computer Science and Control” of the Russian Academy of Sciences, 44-2 Vavilov Str., Moscow 119333, Russia

**Keywords:** 5G, age of information, queueing theory, URLLC, mMTC, quality of service

## Abstract

One of the critical use cases for prospective fifth generation (5G) cellular systems is the delivery of the state of the remote systems to the control center. Such services are relevant for both massive machine-type communications (mMTC) and ultra-reliable low-latency communications (URLLC) services that need to be supported by 5G systems. The recently introduced the age of information (AoI) metric representing the timeliness of the reception of the update at the receiver is nowadays commonly utilized to quantify the performance of such services. However, the metric itself is closely related to the queueing theory, which conventionally requires strict assumptions for analytical tractability. This review paper aims to: (i) identify the gaps between technical wireless systems and queueing models utilized for analysis of the AoI metric; (ii) provide a detailed review of studies that have addressed the AoI metric; and (iii) establish future research challenges in this area. Our major outcome is that the models proposed to date for the AoI performance evaluation and optimization deviate drastically from the technical specifics of modern and future wireless cellular systems, including those proposed for URLLC and mMTC services. Specifically, we identify that the majority of the models considered to date: (i) do not account for service processes of wireless channel that utilize orthogonal frequency division multiple access (OFDMA) technology and are able to serve more than a single packet in a time slot; (ii) neglect the specifics of the multiple access schemes utilized for mMTC communications, specifically, multi-channel random access followed by data transmission; (iii) do not consider special and temporal correlation properties in the set of end systems that may arise naturally in state monitoring applications; and finally, (iv) only few studies have assessed those practical use cases where queuing may happen at more than a single node along the route. Each of these areas requires further advances for performance optimization and integration of modern and future wireless provisioning technologies with mMTC and URLLC services.

## 1. Introduction

The rise of Internet-of-Things (IoT) applications experienced over the last decade has given birth to several technologies supporting the exchange of small amounts of data. Based on the latency constraints and reliability requirements of applications [1], these technologies are classified into massive machine-type communications (mMTC) and ultra-reliable low-latency communications (URLLC) [2,3]. Both services are expected to be supported in 5G systems. While the support of URLLC technologies satisfying the 5G constraints in International Telecommunication Union Radiocommunication (ITU-R) M.2410 has not been reported yet [4], pre-5G mMTC technologies, including Long-Range Wide Area Network (LoRaWAN) [5,6], SigFox, Wireless Fidelity (Wi-Fi) Halow, Extended Coverage–Groupe Special Mobile Internet of Things (EC-GSM-IoT), Long-Term Evolution Machine-Type Communication (LTE-M) [7,8], and fifth-generation (5G)-grade systems such as NarrowBand Internet of Things (NB-IoT) and Digital Enhanced Cordless Telecommunications 2020 (DECT-2020) [9,10,11] have been widely deployed already.

One of the most important use cases for mMTC and URLLC applications is exchanging the status of operational equipment between end devices (ED) or between EDs and a control center [12,13,14]. For example, synchronization between robots performing joint actions on a production line in an industrial environment requires the exchange of time-constrained information [15]. Another striking example is in energy grids, where there is a need for timely delivery of information to the control centers. Operating under upper-layer management protocol suites such as supervisory control and data acquisition systems (SCADA, [16]), EDs utilize always-on Radio Resource Control (RRC)-connected states and are polled centrally over regular time intervals. Such behavior induces the worst possible scenario from a system performance point of view, having batch arrival traffic patterns at the air interfaces that have been designed using purely stochastic arrivals in mind. The required latency for most energy grid applications is much smaller than that required by the 5G system specifications in ITU-R M.2410. Additional relevant applications include remote monitoring systems for automotive networks and video surveillance systems.

Conventionally, the performance of time-sensitive applications has been evaluated using the delay latency as the main metric of interest. However, this metric heavily depends on the traffic load in the network, which in turn is a function of the updated inter-generation time at the source(s). Thus, in 2011, the Age of Information (AoI) metric was proposed to quantify the freshness of available information about the state of remote systems [17]. Conceptually, the AoI metric is an explicit function of the updated inter-generation times at the source and network delay, and presumes that only updates received in a timely manner can reflect the current state of the system. Thus, outdated updates are usually useless. The AoI allows the performance of mMTC and URLLC technologies to be characterized in detail, and can be considered as a measure of the quality of service (QoS).

By explicitly accounting for the packet inter-generation time at the source and the delay associated with packet delivery, the AoI and its maximum term, the peak AoI (PAoI), drastically differ from the delay metric, which is conventionally understood as the interval between the time instant at which the source sends the packet to the network and the instant that the packet is received at the destination. Contrarily, the AoI is defined as the time elapsed since the previous update generation time, i.e., Δ(t) = $t-t_{m}$, where $t_{m}$ is the instant when the most recently received update was generated. From the theoretical point of view, this property makes the AoI analysis harder compared to using the delay. Recalling that the delay is often evaluated using the well-defined time instant at which the packet arrives at the destination; for the AoI, however, it is necessary to additionally account for the random terms of the time instant between *t* and $t_{m}$. These properties force investigators to address simple models in order to account for those having exponential inter-arrival and service times, such as first come first served (FCFS) or last come first served (LCFS) queueing disciplines; see Section 3 for detailed review of this issue.

From the practical perspective, such simple models generally are not able to capture the specifics of the service process of the transmission medium, where packets may arrive in batches, more than a single packet (possibly from multiple sources) can be served in a single frame, the service discipline may depend on the channel states of users, and the service can be provided via multiple phases (such as random access followed by data transmission, as is often the case in cellular mMTC technologies such as LTE-M and NB-IoT). Thus, at present there is a large gap between theoretical investigations and practical applications of the models developed for the AoI, as is discussed in Section 4.

### 1.1. Goals and Contributions

The goals of this paper are to: (i) provide a broad outlook on the current state of queueing-theoretic models for the AoI/PAoI performance assessment of wireless technologies for URLLC and mMTC applications and (ii) identify the gaps between theoretical models and practical applications for mMTC/URLLC services in 5G systems. To this end, we begin with a formal introduction of the AoI and PAoI metrics for typical IoT applications. Then, we proceed to provide a detailed overview and classification of studies performed to date in both theoretical and applied settings. Finally, we discuss in detail the open problems and research challenges related to the design and development of AoI/PAoI-oriented URLLC and mMTC services.

The major contributions of our study are:A detailed analysis of the components contributing to the AoI and PAoI metrics in the general setting of state update applications implemented over a cellular system.An overview of the existing queueing-theoretic and applied studies addressing the AoI and PAoI metrics and identification of gaps between the reviewed wireless technologies and queueing-theoretic models.Outlining of the open problems and research challenges related to the development of the queueing-theoretic tools for performance assessment of the AoI/PAoI metrics for future mMTC and URLLC technologies.

### 1.2. Paper Structure and Other Reviews

We specifically note that there have been several review and tutorial-type papers related to the AoI and PAoI metrics published over the last few years. In [18], the authors discussed the concept of the AoI and the different metrics associated with it. They presented an overview of the first phase of research on the AoI (2011–2016) and discussed future research directions, mainly concentrating on queueing-theoretic tasks. The authors of [19] focused on the state of development of low-latency cyber–physical systems and applications with limited system resources. Their paper presented the AoI-related metrics and general methods for analysing them. In addition to reviewing the existing analytical approaches, the authors considered how the AoI metric is related to methods of sampling, estimation, and control of stochastic processes. In [20], the authors explored Ambient Intelligence (AmI), which represents a future vision of intelligent computing; depending on the relevance of the collected information, it can be measured by the AoI metric. Their paper presented a comprehensive literature review of the AoI in large-scale networks. The authors discussed the impact of queueing policies, stochastic modeling, scheduling, and multiple access schemes. In addition, non-orthogonal multiple access (NOMA), which is considered one of the key technologies outside of 5G, was treated in the context of the AoI.

Compared to the abovementioned studies, in addition to providing a review of the state of the art of theoretical and practical models proposed for the AoI/PAoI metrics, in this paper we attempt to identify the gap between theoretical models and practical applications in the context of mMTC and URLLC services. To this end, we first demonstrate that the notion behind these metrics is more complex than alternatives, involving multiple components in addition to the packet inter-generation time and network delay. In practical systems, these components may significantly affect the resulting AoI/PAoI performance of the system. Further, we show that there is a huge gap between theoretical studies and practical mMTC/URLLC services that does not allow the results derived to date to be utilized for the dimensions of real systems.

The rest of the paper is organized as follows. In Section 2, we identify the components of the AoI/PAoI for a generic state updating service running on top of the cellular system, showing that it may include multiple stochastic and deterministic components that need to be accounted for in addition to the packet inter-arrival time and queuing delay at a single service point. Further, in Section 3 we carefully review the recent theoretic and applied models proposed to date for the AoI/PAoI performance analysis and optimization. Finally, equipped with the information provided in Section 2 and Section 3, in Section 4 we highlight the gaps between modern state update services, including URLLC and enhanced Mobile Broadband (eMBB), and the models proposed to date. Finally, we formulate the critical specifics of future systems models for the AoI/PAoI performance analysis and optimization. Our conclusions are drawn in the last section.

## 2. Age of Information in Technical Systems

In this section, we first introduce a typical system model for a state update IoT service. Then, we proceed to define the notions of the AoI and the PAoI for two cases of interest: (i) single source and (ii) multiple sources.

### 2.1. System Model

As an example, consider the monitoring system for the Industrial IoT (IIoT) illustrated in Figure 1, intended for tasks such as monitoring and control of factory machines. We assume that the machines are equipped with EDs having dual functionality: (i) regular measurement of a certain metric of interest related to the factory machine (temperature, humidity, vibration, etc.) and (ii) transmitting the collected data. The EDs are wirelessly connected to the remote control center (CC) via a base station (BS); the CC utilizes this information to (a) analyse the dynamics of each machine over a certain long period for diagnostic purposes [21] and (b) in the case of an alarming state, make a decision and send a control command to the machine to promptly change its mode of operation [22]. Note that the CC can be installed as an edge service [23,24,25] in the 5G network or be located on the internet.

We utilize Figure 1 to demonstrate a typical example of a remote monitoring system in which many end devices (ED) are connected to the remote control center via wireless technology (e.g., 5G cellular). In terms of 5G cellular systems, the architecture of which is described in TS 23.501 [26], the base station (BS) is a gNB, while the gateway (GW) is the user plane function (UPF) termination. The control center (CC) can be implemented as a part of the 5G edge services or located on the internet. In our case, due to the nature of the service, namely, real-time state monitoring, we utilize the former option.

Note that in case (a) the CC is interested in all the information received from the EDs, while in case (b) the CC is only interested in the latest am=nd most accurate information from each ED, the so-called informational packets [27]. The latter use case is often considered in the literature [28,29]. At the same time, in the former case, less fresh non-informative [27] packets can be delivered to the destination. There are several applications of these use cases, such as digital twins [30,31], i.e., replicas of complex operational systems and timely control for simple monitoring. However, for all of these applications the information in the CC must be updated promptly, implying that the goal of the system is to minimize the time of update delivery from the source ED to the CC.

The model of the considered system (see Figure 1) includes a finite number *N* of EDs, a wireless communication channel $link_{1}$, a BS with a polling-based scheduler for the EDs, a high-speed communication channel $link_{2}$ in the operator’s core network, a gateway (GW) to the internet through the operator’s network, a high-speed $link_{3}$ communication channel with the internet, a control center (CC) with a Control Function (CF), and a database for storing the information from the EDs received in the update packets (which we henceforth refer to as “updates” for brevity). Here, by a CC we refer to an element of a feedback control loop consisting of a monitor, a control function, and an actuator. The monitor located in the CC takes care of the database; depending on the monitor readings, the CF determines whether and what kind of control actions are needed and sends a command to the actuator; finally, the actuator executes the received command. For example, for an actuator located in the factory, based on the command, the operating mode of the factory machine or the interval between measurements, and generation of updates on the ED side may be changed. Simultaneously, the command may include the inputs for the scheduler algorithm R to set the order for polling the EDs. The transmission delay at the $link_{1}$ channel includes the time to serialize the packet to the channel, the propagation time of the radio signal in the transmission medium, and the time at which the update is received by the BS. The radio resource allocation (RRA) scheduler located at the BS, determines which EDs are to be polled in order to perform measurement, then sends the updates within the scheduled time interval. The BS processes the received update and stores it temporarily until it is selected by the scheduler for transmission over the $link_{2}$ channel to the GW and serializing to the $link_{2}$ channel. The model takes into account the time required for the update to be processed by the GW, further transmission over the $link_{3}$ channel, and update processing on the CC side, which includes appending the update to the CC database and the operation of the CF.

One of the goals of investigating this system is to develop an scheduler algorithm R that is efficient in terms of both the polling order and the rule for selecting updates queued at the EDs for further transmission to the CC. For this purpose, the following information available on the CC side after processing the update is useful: the time instant of measurement of the ED readings, which may coincide with the instant of update generation; a set of readings of the ED, for example, temperature, humidity and/or vibration, and other information; and the time instant of the end of appending the ED readings to the CC database. The information can be used by the CF to control scheduling on the BS side, the intensity of update generation on the ED side, or for other purposes. Let each update be stored in the CC log database as a tuple of four parameters:*i*—the ED index;$t^{\left(i\right)}$—the timestamp corresponding to the time instant of update generating on ED *i* side;$r^{\left(i\right)}$—the set of readings (e.g., $r_{1}^{\left(i\right)}$—temperature, $r_{2}^{\left(i\right)}$—humidity, $r_{3}^{\left(i\right)}$—vibration, etc.) contained in the update generated at time instant $t^{\left(i\right)}$ by ED *i*;$t^{\prime (i)}$—the timestamp corresponding to the time instant of the end of update processing on the CC side.

The tuple $\langle i,t^{\left(i\right)},r^{\left(i\right)},t^{\prime (i)}\rangle$ for the ED *i* allows the monitor on the CC side to evaluate the obsolescence of the data contained in the third parameter $r^{\left(i\right)}$ at any time instant by utilizing the timestamp (i.e., the second parameter $t^{\left(i\right)}$ of the tuple). The fourth parameter $t^{\prime (i)}$ can be involved to form initial data for the efficient operation of the scheduler algorithm R.

Note that in case (a) the CC database should store all the tuples received during the monitoring interval, while in case (b) only the last tuple, which contains the set of the most recent readings received by the CC as well as the time instants when the ED has performed measurements, is stored. Therefore, in case (b) it is enough to preempt the previous data from this ED stored in the CC database and replace these with the fresh data every time an update from this ED is received on the CC side.

### 2.2. Single ED System

To introduce the definition of the AoI and the PAoI, in this subsection we begin with the special case of a single ED omitting the index of EDs *i*. The freshness of any information stored in the CC database reading $r_{k}$ for the *k*-th physical parameter of the ED is determined by the time interval that has elapsed since the instant of this reading’s measurement.

Figure 2 shows the detailed time diagram from the instant of the reading’s measurement until the end of handling the corresponding update packet on the CC side. The notation we utilize for this system is presented in Table 1.

#### 2.2.1. System Operation

The AoI is a function of time [32], which is a measure of the obsolescence of information from the ED available at the CC. Note that in case (a) introduced in Section 2.1 it is possible to determine the freshness of any reading received on the CC side, as all tuples are stored in the database, while in case (b) it is only possible to determine the freshness of the last tuple received at the CC side.

To define the AoI, let us consider Figure 2 and concentrate on one of the reading types, i.e., humidity ($r_{2}$). Considering all the components of the time interval from the instant of the reading’s measurement until the end of handling the corresponding update packet on the CC side including CF execution, according to this diagram, the reading $r_{2}$ measured on the interval [$t_{n-1},t_{n}$) is handled on the the CC side through time $\xi_{2}+T_{n}$ after the instant of measurement. The first term concerns the processing of the reading on the source side, while the second involves the transmission delays along the multi-hop route from the source to the destination, including processing on the destination side. The value $\xi_{2}+T_{n}$ determines the obsolescence of the $r_{2}$ reading on humidity at the end of the *n*-th update’s processing on the CC side at the time instant $t_{n}^{\prime }$. After that reading, $r_{2}$ becomes obsolete with the speed of passing time, and at time instant $t_{n+1}^{\prime }$ the reading $r_{2}$ is replaced by a newly arrived humidity measurement.

Note that in Figure 2 several readings (temperature, humidity, vibration) are measured over the interval [$t_{n-1},t_{n}$), all of which are transmitted in a single update packet generated at time instant $t_{n}$. Thus, when estimating the term $\xi_{2}$, two approaches should be considered: (i) for the transmission of each reading, an individual update packet is generated, then $\xi_{2}$ coincides with the updated inter-generating time, which is fixed or adjusted by control commands from the CF; and (ii) several readings are transmitted in a common update packet, then the upper bound, i.e., the update inter-generating time, is taken to estimate the interval from the measurement instant to the generation of the common packet instant $t_{n}$ while assuming that $\xi_{1}=\xi_{2}=\xi_{3}=t_{n+1}-t_{n}$. Thus, both approaches rely on the updated distribution of the inter-generating time $Y_{n}=t_{n+1}-t_{n}$ when estimating the first term $\xi_{2}$ in the AoI.

We now continue with the distribution of the second term, $T_{n}=\sum_{k=4}^{12}\xi_{k}$. Depending on the goal of analyzing the AoI, only certain terms of this sum need to studied in detail, while simplifying assumptions can be made for the rest.

An update *n* at time instant $t_{n}$ starts to be issued to the $link_{1}$, then is transmitted over it and received at the BS side during the interval $d=\xi_{4}+\xi_{5}+\xi_{6}$. The propagation time $\xi_{5}$ over the $link_{1}$ varies for different EDs, and depends on the radio channel conditions (primarily the distance between the ED and the BS, as well as the signal propagation specifics such as fading, line-of-sight blockage, etc.), the organization of the multiple access scheme (e.g., random access, centralized BS control), the resource organization (time/frequency or orthogonal frequency division multiplexing), and the type of BS scheduler. Note that in a multi-source system the parameter *d* is taken into account by the BS schedulers in the RRA procedure. Upon reception on the BS side, the tuple from the update is recorded in the BS storage memory and temporarily stored there during $\xi_{7}$ until the update is selected by the scheduler algorithm R for transmission over the $link_{2}$ towards the CC.

As shown in Figure 1, the model takes into account the time $\xi_{9}$ required for the update to be processed by the GW and the transmission time $\xi_{10}$ over the $link_{3}$. Upon reception at the CC side, the update is recorded to the database during $\xi_{11}$. In addition, we include the CF execution time $\xi_{12}$ in the AoI, as the fresh information from the recent update, in particular the timestamp, can be used by the CF to adjust the intensity of update generation at the ED side and/or as inputs for the scheduler algorithm R at the BS side. The values of the received readings can provoke certain actions to be performed on the ED side as a response to critical readings values. The time instant $t_{n}^{\prime }$ when the information from the *n*-th update has been handled by the CC is important for the AoI and the PAoI; thus, the time instant $t_{n}^{\prime }$ should be recorded in the database of the CC as part of the *n*-th update tuple. During the time interval ($t_{n}^{\prime }$,$t_{n+1}^{\prime }$) the data r from the *n*-th update are considered the most recent, and the AoI is determined by instant $t_{n}$ of the *n*-th update generation.

An example of the CC database structure is shown in Figure 3. The PAoI and the AoI columns are optional, as they can be calculated at any time through the elements of the tuple according to (Equation 1) and (Equation 4) provided below. Note that the CC database recordings can be utilized for analysing the dynamics of the ED in the long run for diagnostic purposes or for making decisions and signalling to promptly change EDs’ mode of operation.

#### 2.2.2. AoI/PAoI Definitions

We now proceed with the formal definitions of the AoI Δ(t) at an arbitrary time instant *t*, the PAoI $A_{n}$ for update *n*, and the PAoI A(t), which is the current PAoI at time instant *t*. Note that all three metrics are determined on the CC side.

**Definition** **1.**
*The AoI Δ(t) of an ED is a function of the time [18]; for an arbitrary time instant t, it is defined as*

(1)
$$\Delta \left(t\right)=t-t_{m},$$

*where*

(2)
$$m=sup\{n:t_{n}^{\prime }\leq t\},\;n\geq 1,\;\;m\geq 1.$$



**Definition** **2.**
*The PAoI $A_{n}$ of an ED is a function of the update index n, depends on the generation time instant for update (n−1) and on the processing time instant for update n already recorded in the CC database, and is determined as follows:*

(3)
$$A_{n}=\Delta \left(t_{n}^{\prime }-0\right)=t_{n}^{\prime }-t_{n-1},\;n\geq 1.$$



Because the PAoI $A_{n}$ of an ED for update *n* is expressed in terms of the AoI Δ(t), the dependence of the current PAoI on the time instant *t* when entered the CC database can be written as follows: (4)$$A\left(t\right)=\Delta \left(t_{m}^{\prime }-0\right)=t_{m}^{\prime }-t_{m-1},$$
where *m* is defined in (Equation 2).

Note that the functions AoI Δ(t) and PAoI A(t) can be interpreted as continuous-time continuous state processes over the state space R (see Figure 4). As can be seen, the PAoI process A(t) is a step-wise random process with jumps that occur at the end of the update’s handling on the CC side. The state of A(t) does not change until update (n+1) is handled. Thus, the PAoI A(t) can be described as a continuous time process over the same state space R. The corresponding discrete-time process $A_{n}$ is an embedding of Δ(t) at the time instant just before $t_{n}^{\prime }$.

#### 2.2.3. Detailed AoI/PAoI Dynamics

We now consider the dynamics of the AoI/PAoI metrics in Figure 4 and Figure 5 in detail. We observe Δ(t), $A_{n}$, and A(t) at arbitrary time instants $t=l_{1}$, $t=l_{2}$, and $t=l_{3}$, as illustrated in Figure 5.

Observe that the transmission of the first bit of update *n* starts at time instant $t_{n}$, while at time instant $t_{n}^{\prime }$ the processing of update *n* on the CC side is fully completed. Consider the system at an arbitrary time instant $l_{1}$. By this time instant, the last processed (i.e., recorded in the CC database) update in the system has the index *n*; thus, from (Equation 3), the PAoI $A_{n}=t_{n}^{\prime }-t_{n-1}$. Therefore, the value of the AoI at time instant $l_{1}$ is $\Delta \left(t\right)=l_{1}-t_{n}$, and the value of the PAoI is $A\left(l_{1}\right)=t_{n}^{\prime }-t_{n-1}$, i.e., $A\left(l_{1}\right)=A_{n}$.

Then, at time instant $t_{n+1}$, the transmission of the first bit of the update with index (n+1) begins. Even though by the time instant $l_{2}$, $t_{n+1}<l_{2}\leq t_{n+1}^{\prime }$, two updates *n* and (n+1) have already been generated at the ED, only the first of these has been processed at the CC side; thus, from (Equation 3), the PAoI remains $A_{n}=t_{n}^{\prime }-t_{n-1}$. Therefore, the value of the AoI at time instant $l_{2}$, which is determined on the CC side, continues to depend on the same update with index *n*; at time instant $l_{2}$, the value of the AoI is $\Delta \left(l_{2}\right)=l_{2}-t_{n}$ and the value of the PAoI is $A\left(l_{2}\right)=t_{n}^{\prime }-t_{n-1}=A_{n}$, that is, $\Delta \left(l_{2}\right)>\Delta \left(l_{1}\right)$, $A\left(l_{2}\right)=A\left(l_{1}\right)$.

At time instant $t_{n+2}$, the transmission of the first bit of update (n+2) is initiated. Further, the processing of update (n+1) is completed at time instant $t_{n+1}^{\prime }$, while update (n+2) is finalized at time $t_{n+2}^{\prime }$, resulting in the PAoI $A_{n+2}=t_{n+2}^{\prime }-t_{n+1}$. Thus, at $l_{3}$ the CC has already processed update (n+2); therefore, the AoI value is updated, and from (Equation 4) we have the AoI $\Delta \left(l_{3}\right)=l_{3}-t_{n+2}$, while from (Equation 3) we have the PAoI $A\left(l_{3}\right)=t_{n+2}^{\prime }-t_{n+1}$, i.e., $A\left(l_{3}\right)=A_{n+2}$.

In addition, note that if $T_{n}=t_{n}^{\prime }-t_{n}$ is the end-to-end delay of update *n* from the ED to the CC and if $Y_{n}=t_{n}-t_{n-1}$ is the update inter-generation time between updates (n−1) and *n*, then the AoI Δ(t) at an arbitrary instant *t* can be defined as $A_{n}=\;T_{n}+t-t_{n}^{\prime }$, while the PAoI $A_{n}$ arrived at just before receiving the *n*-th update can be defined as [33] $A_{n}=\;Y_{n}+T_{n}$. The last expression is utilized to determine the AoI/PAoI in both practical and theoretical models.

### 2.3. Multiple EDs

We now consider a system with *N*, N>1, and multiple EDs. Before formally defining the AoI and PAoI metrics for this case, we modify the notation in Table 1 by utilizing the index *i* to denote the ED numbers, i.e., i=1,…,N. The definitions for the case with multiple EDs follow directly from (Equation 1)–(Equation 4).

**Definition** **3.**
*The AoI $\Delta^{\left(i\right)}\left(t\right)$ of ED i depends on the time instant $t_{m}^{\prime (i)}$ when the last update received before t from ED i was handled on the CC side:*

(5)
$$\Delta^{\left(i\right)}\left(t\right)=t-t_{m}^{\left(i\right)},$$

*where*

(6)
$$m=sup\{n:t_{n}^{\prime (i)}\leq t\},\;n\geq 1,\;\;m\geq 1,\;\;i=1,\ldots ,\;N.$$



**Definition** **4.**
*The PAoI $A_{n}^{\left(i\right)}$ of ED i depends on the sequence number n of the update from this ED recorded in the CC database, and is defined as*

(7)
$$\;A_{n}^{\left(i\right)}=\Delta^{\left(i\right)}\left(t_{n}^{\prime (i)}-0\right)=t_{n}^{\prime (i)}-t_{n-1}^{\left(i\right)},\;n\geq 1,\;\;i=1,\ldots ,\;N.$$



Because the PAoI $A_{n}^{\left(i\right)}$ for the *n*-th update from ED *i* is expressed in terms of $\Delta^{\left(i\right)}\left(t\right)$, its dependence on the time instant $t_{m}^{\prime (i)}$ when it was handled on the CC side can be written as
(8)$$A^{\left(i\right)}\left(t\right)=\Delta^{\left(i\right)}\left(t_{m}^{\prime (i)}-0\right)=t_{m}^{\prime (i)}-t_{m-1}^{\left(i\right)}$$
where *m* is defined in (Equation 6).

Returning to the Figure 2, note that in the case with multiple EDs the individual distribution of the random variable $\xi_{5}$ representing the propagation delay, which depends on the RRA scheduler located at the BS, should be taken into account for the ED. The presence of multiple EDs significantly complicates the availability of an efficient scheduler algorithm R, both in terms of the order in which EDs are polled and in terms of the rule used to select updates queued at the EDs for further transmission to the CC.

Thus, for systems with multiple EDs operating in stationary mode, a multi-objective optimization problem can be formulated with one of the multi-parameter objective functions based on the time average, ensemble average, or both. In case (a) introduced in Section 2.1, a solution to this problem can be obtained based on the statistics collected over some relatively large observation period *T*, which provides the desired algorithm R for the scheduler. An example of such problems in the case of multiple devices is
(9)$$\begin{array}{c} E\Delta^{\left(i\right)}\left(\left[t,t+T\right]\right)\to \min,\;\;s.t.\;N,\;T,\;R{,}^{\left(i\right)} \end{array}$$
or alternatively,
(10)$$\begin{array}{c} \mathrm{Max}A^{\left(i\right)}\left(n,\left[t,t+T\right]\right)\to \min,\;\;s.t.\;N,\;T,\;R. \end{array}$$

Here,
(11)$$E\Delta^{\left(i\right)}\left(\left[t,t+T\right]\right)=\int_{t}^{t+T}\Delta^{\left(i\right)}\left(t\right)dt,\;\;i=1,\ldots ,\;N$$
is the time the average AoI for ED *i* during the observation period $\left[t,\;t+T\right]$, and
(12)$$\mathrm{Max}A^{\left(i\right)}\left(n,\left[t,t+T\right]\right)=\max_{n=n_{min},\ldots ,n_{max}}A_{n}^{\left(i\right)},\;i=1,\ldots ,N$$
is the maximum PAoI for ED *i* during the observation period $\left[t,\;t+T\right]$, which is the longest delay in receiving an update from ED *i* during the observation period, where
(13)$$n_{min}=\inf\left\{n:\;t_{n}^{\left(i\right)}\geq t,\;n\geq 1\right\},\;\;n_{max}=\sup\left\{n:\;t_{n}^{\prime (i)}\leq t+T,\;n\geq 1\right\}.$$

For systems operating in non-stationary mode, the optimization problem practically becomes a real-time problem, and as such there is no need to investigate the time average. Such problems arise in case (b) described in Section 2.1. An example of objective functions can be the ensemble average value of the current AoI at time instant *t*:(14)$$E\Delta \left(t\right)=\frac{1}{N}\sum_{i=1}^{N}\Delta^{\left(i\right)}\left(t\right),$$
or the ensemble maximum value of the current PAoI value at time instant *t*: (15)$$\mathrm{Max}A_{n}\left(t\right)=\max_{i=1,\ldots ,N}A_{n}^{\left(i\right)}\left(t\right),$$
i.e., the PAoI from the ED with the oldest update among all EDs received by the CC. The solution to the problem can determine the algorithm used for ED polling, the rule for selecting the packet stored in the queue on the BS, or the recommended frequency for generating updates, which, as a result of the operation of the CF, should be transmitted to the BS and EDs. Examples of objective functions in a similar optimization problem include the information value of the update (VoIU), the cost of update delay (CoUD), and the peak cost of update delay (PCoUD) [34].

### 2.4. AoI/PAoI Analysis Challenges

It is important to observe that both the AoI and the PAoI are random variables. For their analysis, it is necessary to know the exact or approximate probability distribution for each of the random variables $\xi_{1}-\xi_{12}$, which in general complicates the analysis.

The most complex part of the AoI/PAoI analysis is specifying the components of the update delay in the nodes of the route, namely, $\xi_{6},\xi_{7},\xi_{9},\xi_{11},\xi_{12}$. In [27], this problem was solved using the apparatus of queueing theory, while the update delay in a node was modeled as the sojourn time of the request in a GI/GI/1 queueing system with a general distribution of both the inter-arriving time and the service duration. The Laplace–Stieltjes transform of the PAoI distribution function can then be obtained for two essentially different possible cases of mutual placement of the update’s generation point and end of the update’s processing point. The difference is visible in Figure 5, where a case $A_{n-1}=S_{n}$ corresponds to $t_{n-2}^{\prime }<t_{n-1}$ and a case $A_{n}=W_{n}+S_{n}$ corresponds to $t_{n-1}^{\prime }>t_{n}$, with $W_{n}$ and $S_{n}$ being the corresponding wait times and service durations of the request in the queueing system.

Although the results of [27] are valid for the wide class of distributions, the difficulty lies in determining the form of the approximate distribution of the inter-arrival time and service duration. For example, while simulating the update delay on a BS for the case of a single source, the distribution of the inter-arriving time depends on the propagation time $\xi_{5}$, which is a random variable depending on many factors, as described above in Section 2.2.1. According to [27], the aggregated flow from all sources should be considered as an input when addressing the case of multiple sources.

In addition, for wireless channels with high bandwidth allowing reception of updates from several sources, queueing systems with group arrival would be more adequate. In a case with high bandwidth links, the BS scheduler may select multiple updates for simultaneous transmission to the GW, which can be modeled by a queueing system with group service. One more possible queueing system apparatus to simulate the process of collecting updates from end devices is a polling system. Thus, to build a more adequate reality of the model, a deeper apparatus of queueing theory than the classical one can be used.

### 2.5. AoI/PAoI Examples

We now proceed with two simple examples that demonstrate the nature of the AoI/PAoI metrics and compare them to the classic time-related metric of the full delay experienced in the system. To this end, we consider two queuing models M/M/1 and M/D/1, which, as shown in Section 3.1 and Section 3.2, have been well studied in the theoretical setting and applied in practice [35,36]. Let the update generation process be homogeneous Poisson in nature, with rate λ; the service time is either exponential or deterministic, respectively, with the mean $\mu^{-1}$. To ensure the stability of the queueing process, we consider that the server utilization ρ, defined as λ/μ, is smaller than unity. We are interested in the mean PAoI and the mean full delay (sojourn time).

For both systems, in the stationary regime, the mean full delay experienced by an update in the system can be written as
(16)$$\begin{array}{c} T=\omega +\mu^{-1} \end{array}$$
while the mean waiting time in the buffer obeys the Pollaczek–Khinchine formula [37].
(17)$$\begin{array}{c} \omega =\frac{\lambda^{2}V^{\left(2\right)}}{2(1-\rho )} \end{array}$$

The second moment $V^{\left(2\right)}$ of the update service time is provided by
(18)$$\begin{array}{c} V_{exp}^{\left(2\right)}=\frac{2!}{\mu^{2}},\;V_{det}^{\left(2\right)}=\frac{1}{\mu^{2}}, \end{array}$$
where the indices “exp” and “det” refer to M/M/1 and M/D/1, respectively.

Then, the mean delay in the buffer can be calculated as
(19)$$\begin{array}{c} \omega_{exp}=\frac{\rho^{2}}{1-\rho },\;\omega_{det}=\frac{\rho^{2}}{2(1-\rho )}. \end{array}$$

Therefore, the mean PAoI in the system can be estimated as follows:(20)$$\begin{array}{c} A_{exp}=\frac{\rho^{2}}{1-\rho }+\mu^{-1}+\lambda^{-1},\;A_{det}=\frac{\rho^{2}}{2(1-\rho )}+\mu^{-1}+\lambda^{-1}, \end{array}$$
where the first part is responsible for the full delay and the second $\lambda^{-1}$ is responsible for the intensity of update generation at the source.

Figure 6 and Figure 7 demonstrate the behavior of the calculated PAoI and corresponding full delays in the considered systems as a function of ρ. As can be observed, the mean PAoI metric is extremely large for small values of ρ. The rationale is that updates are rarely generated at these values, and although they experience almost no queuing delay (see Figure 7), the second part of the $A_{exp}$ and $A_{det}$ dominate leading to the large values of the mean PAoI. As ρ starts to increase, the mean PAoI quickly drops, as the second component in $A_{exp}$ and $A_{det}$ vanishes more quickly compared to the increase in the mean full delay. It cn be observed that the mean PAoI eventually attains its minimum. Notably, up until that point both systems behave almost identically from the point of view of the mean full delay and the mean PAoI. However, as ρ increases even further, the full delay component, which is different for the considered systems, starts to dominate. For both systems, it can be seen that the mean PAoI increases quickly as the systems become overloaded, leading to a full delay in specifying the behavior of the metric. As M/D/1 is generally characterized by smaller full delay due to deterministic service times [37], the rise of the mean PAoI is slightly delayed for this system compared to M/M/1. Note that this behavior is typical for queuing systems with FCFS service discipline.

## 3. The State of the Art in AoI and PAoI Models

The concept of the AoI was originally introduced in 2011 by Kaul [17] to quantify the freshness of knowledge about the state of a remote system, and immediately spawned interest from both theoretical and practical points of view. Practical interest is due to the rising number of applications that exchange state updates which need to be supported by URLLC and mMTC 5G services. On the other hand, the AoI is a novel metric for queueing theory, drawing purely theoretical interest as well. Specifically, at the beginning of the 2010s researchers mainly concentrated on evaluating AoI performance in various technical systems using computer simulations. Lately, the AoI has begun to attract attention from the queueing systems community as well, and nowadays these two areas are being developed in parallel.

Recall that, under the first come first served (FCFS) service discipline, the AoI and PAoI metrics increase at both low and high state sampling rates. This is the main property that distinguishes them from the delay, which has been conventionally utilized in queueing theory, and is an increasing function of the sampling rate. Even though the AoI and the PAoI are gaining popularity, they remain insufficiently well studied from both practical and theoretical perspectives, as we discuss below.

Based on the discussion above, we divide all of the reviewed studies into the two main categories of practical and theoretical. The former group encompasses models of technical systems, such as 4G, 5G, and Institute of Electrical and Electronics Engineers (IEEE) cellular and wireless local area networks (WLAN) as well as services such as URLLC and mMTC. In this approach, the formulated model is studied using either computer simulations or queueing-theoretic tools. Theoretical models are mainly inspired by the white spots in queueing theory, and sometimes are not related to practical systems. Table 2 presents a summary of the reviewed studies.

### 3.1. Queueing-Theoretic Studies

Most theoretical studies consider single-server systems in either discrete time or continuous time. Early AoI studies often concentrated on simple models in terms of the number of sources, arrival and service processes, number of servers, and queueing discipline. Specifically, in [28], three simple queueing models M/M/1, M/D/1, and D/M/1 were studied within the framework of the FCFS discipline. The authors addressed the problem of determining the link rate such that the duration between state updates is the upper bound at the destination. This work introduced the AoI metric, which was averaged over time to evaluate the performance of state update services. For the considered queueing systems, the authors demonstrated that there is an optimal inter-generation update rate at the source that minimizes the mean PAoI. This rate is different from that used for maximizing the system’s throughput or minimizing the latency. Specifically, it has been shown that the minimal mean PAoI is achieved when a new packet is available exactly when the previous packet completes its service.

The studies in [38,39] assumed a Poisson arrival process as input and considered queueing systems of $\sum_{i}$M/M/1 types with multiple independent sources providing the state updates to a common destination and FCFS service discipline. Note that the ensemble average AoI over all the sources is rarely considered in studies addressing multiple sources. Instead, a tagged source is often chosen and time-averaged AoI characteristics are derived. Further, in [38] Kaul dropped the FCFS assumption by allowing newly generated status updates to precede the old ones in the queue, and assumed LCFS discipline. This simple queue management mechanism improves system performance with respect to the AoI metric. The LCFS service discipline has subsequently been adopted in other practical studies; see, e.g., [40].

In [49], it was shown that allowing preemption during service leads to a lower average AoI in the system. Here, the authors considered a multi-source state update system, where each source generates state update packets according to a Poisson process and each update requires normally-distributed service times. The resulting system in Kendall’s notation with multiple sources is then $\sum_{i}$M/G/1/1. Three types of policies have been considered: (i) no preemption; (ii) self-preemption, where the newest packets from the same source may interrupt the older ones; and (iii) source-independent preemption, where a new arriving packet may preempt the older packet in the system regardless of its source. The Moment Generating Function (MGF) of the AoI has been obtained for all of the considered policies. Numerical results have shown that each policy can outperform others depending on system parameters, i.e., packet arrival rate and service time distribution.

Systems with multiple sources and specific types of packet management policies have been considered as well. Specifically, in [58] the authors assumed the packets of each source to be generated according to the Poisson process and the packets to be served according to an exponentially distributed service time. The average AoI of each source was of interest. The authors obtained this using the Stochastic Hybrid Systems (SHS) method for each package management policy they considered. In particular, the policy allowing for no more than two packets from the same source in the queue was considered. If the server is busy when a packet arrives, a packet from the same source waiting in the queue is replaced by a new packet that has arrived. The second studied policy was that the system could contain no more than two packets and at most one from each source. If the server is busy when a packet arrives, then a packet from the same source in the system is replaced by a fresh packet. Finally, the third policy was similar to the second except that it did not allow for preemption while the older packet was in service. Numerical results showed that the second policy leads to a lower total average AoI in the system as compared to other policies, with the third policy providing the fairest AoI among the sources.

The study in [39] was the first to address generally distributed service times, a topic that is of special interest in practical applications such as cellular systems and multi-hop wireless networks. The authors of [40] further considered the Gamma distribution of service times with LCFS service discipline. In the former study, a special case of Gamma distribution was considered, namely, Erlang distribution, which captures the process of packet retransmissions over multiple relay nodes. Both papers addressed service with and without preemption. The analytical results obtained by these authors coincided with the simulation results. The comparison of the performance of the considered schemes demonstrated that the strategy without preemption minimizes the mean AoI for Gamma-distributed service times. The authors additionally observed that this strategy remains optimal when deterministic service times are considered.

The M/G/1/1 queueing system with preemption, no buffer space, and multiple sources was considered in [43]. The authors derived the AoI utilizing the bypass flow graph method, and showed that it is impossible to prioritize one of the flows with a fixed total arrival rate while simultaneously minimizing the total AoI. To minimize the total AoI, all the sources must generate packets at the same rate, implying that no individual stream can be assigned a higher rate, which is a necessary condition for reducing the AoI as compared to other streams.

Note that one may utilize advanced packet scheduling strategies to avoid network congestion and minimize the AoI. Specifically, one may increase the efficiency of the system by discarding or replacing packets. The studies in [32,41] addressed this topic and demonstrated that there are scheduling strategies that allow for minimizing AoI as compared to simple queueing disciplines such as FCFS or LCFS. In [32], the authors proposed a general methodology called “stochastic hybrid systems” for AoI that applies to a wide range of queueing systems with multiple sources. For systems with FCFS and LCFS service disciplines, Poisson arrivals, and exponential service times, the range of possible average AoIs has been explored in detail.

The PAoI/AoI cumulative distribution functions (CDF) were addressed in [27], where the authors presented a general expression for the CDF of the AoI for a general class of state update systems. Specifically, they showed that the AoI distribution can be obtained via the delay CDF and the peak AoI CDF. Next, the authors derived the AoI CDF for a large class of queueing systems with different service disciplines, including FCFS and LCFS.

The PAoI metric providing the upper bound for the AoI is of considerable interest to the scientific community. This metric is generally simpler to derive; thus, in addition to the mean values, distributions of the PAoI have been reported. In [60], the authors calculated the PAoI using the example of technical system operating on the basis of the IIoT. They explained how this parameter changes when a new update is generated in the system, and demonstrated this using a graph. In [34], the authors considered discrete-time Geo/Geo/1 and Geo/Geo/1/1 queueing systems with FCFS and LCFS service disciplines and preemption. The authors derived the CDF and probability mass function (PMF) for both the AoI and PAoI metrics. Note that, as compared to the mean values, distributions can be utilized to provide performance guarantees in terms of the percentiles/quantiles of the AoI and the PAoI. The results of the work can be extended to obtain closed-form expressions for the mean AoI. For this, the representation of CDF in the form of a power series is used. The same paper presented a general formula for the stationary distribution of the AoI which is valid for a wide class of state update systems in discrete time. In [52], the authors investigated the PAoI and average AoI for similar queueing systems. Specifically, they analysed Geo/G/1 and Geo/G/1 with FCFS service discipline.

Discrete-time systems have been considered by other authors as well; in [59], the authors studied a discrete-time queue arising in multi-source IoT systems. Assuming geometrical updating of the inter-generation time, the exact distributions of the AoI and PAoI were numerically obtained using the matrix-geometric approach for three different settings: (i) non-preemptive bufferless (NPB), (ii) preemptive bufferless (PB), and (iii) non-preemptive single buffer with replacement (NPSBR). Numerical examples have shown that the PB discipline provides the best performance in most cases; however, it is slightly inferior to the NPSBR discipline when the average service rate is small.

The AoI and PAoI metrics are essentially end-to-end, meaning that the packets can be queued at multiple intermediate servicing systems. Many authors have attempted to explicitly reduce this to a single bottleneck system. In [48], the problem of minimizing AoI in wireless networks was considered under general interference constraints and time-varying channels in a single-hop wireless network. The authors studied two different scheduling policies. Specifically, they proposed virtual queue-based and explicit AoI-based scheduling policies, where the scheduler uses the information about the current state of the link to minimize the AoI. As a result, the policy based on the virtual queue turned out to be optimal up to a constant additive coefficient, while the policy based on the AoI may deviate from the optimal values by more than four times. In [50], the authors explicitly addressed the case of a multi-hop network. In [51], the authors presented the first study proposing a scheduling policy that optimizes the AoI in a wireless network with unreliable channels.

At the end of our review of theoretical studies, we note several topics that have received only a little attention to date. These include systems with multiple sources having different arrival rates, complex inter-arrival and service times, queueing networks, etc. In [55], the authors studied a probabilistic preemptive bufferless queue of M/PH/1/1 type under load from multiple sources with self-preemptive discipline. As compared to [39], in this scenario the service time distribution parameters for the sources are different. They derived the distributions of the AoI and the PAoI for each of the sources, then demonstrated that it is possible to minimize a class of AoI-oriented utility functions. In [56], queueing models with one server and multiple sources were considered, with each source independently generating state update packets according to the Poisson process. The AoI was estimated for each source. In addition, the authors obtained approximate expressions for the average AoI in a system that has a common service time distribution. In [57], the authors showed that bufferless and single-buffer queueing systems can effectively cope with the increased AoI that occurs in queueing systems with FCFS service discipline. In their paper, a numerical algorithm was proposed to obtain the exact distributions of the AoI and PAoI in a bufferless PH/PH/1/1/P(p) queueing system with probabilistic preemption and in a single-buffer M/PH/1/2/R(r) queue with probabilistic packet replacement. The resulting distributions have a matrix exponential form, which makes it possible to directly calculate the tail probabilities.

### 3.2. Applied Studies

The AoI occupies an important place in many applied studies where researchers have considered real technical systems and service provision. In [17], the authors utilized the M/M/1 model with infinite buffers and considered applications that require network nodes to periodically share their time-critical state information with neighboring nodes or BS. Automotive networks are a good example of such an application, where each vehicle shares its data (for example, its location) with neighboring vehicles to improve road safety. They assumed a broadcast service as the information distribution mechanism. The authors had the goal of studying the AoI metric when maintaining the information about the current state of the traffic at all nearby nodes. As the technical system of interest, they investigated the IEEE 802.11p Carrier Sense Multiple Access (CSMA)-like medium access control (MAC) mechanism and compared various service disciplines to optimize the AoI. They demonstrated the fundamental difference between minimizing the system throughput and AoI. By utilizing computer simulations for a M/M/1 queueing system, they minimized the AoI to the optimal load point, which is located between the points maximizing the system throughput and minimizing the delay. In addition, simulations have shown that this optimal point cannot be achieved in IEEE 802.11p systems with CSMA-like MAC.

In applied studies, authors often rely on well-studied queueing systems to model the AoI in real technical systems. In [42], based on the results reported in [43], the authors utilized an M/G/1/1 system with self-preemptive service discipline, where updates that are currently in the system are discarded upon arrival of a new update from the same source. In [42], this model was related to scenarios in which updates are sent over an erasure binary channel using either an infinite incremental redundancy (IIR) hybrid automatic repeat request (HARQ) system or a fixed redundancy (FR) HARQ system. The authors obtained the mean AoI for both systems. The best strategy turned out to be to prioritize the update currently being sent, rather than to preempt it. Moreover, as expected, the IIR HARQ protocol provides better performance in terms of the mean AoI than FR HARQ.

Similarly to theoretical studies, an upper bound of AoI–PAoI, introduced in [28], is often addressed in applied works. The rationale is that PAoI can be utilized in practical scenarios to impose AoI thresholds and provide performance guarantees. For example, in [44] the authors considered heterogeneous systems in which objects generate state messages having different service time distributions. In this context, the authors set themselves the task of obtaining the PAoI for a class of $\sum_{i}$M/G/1 queueing systems with multiple arrival flows. As a result of their study, exact expressions for CDF and probability distribution function (PDF) of the PAoI were obtained. The PAoI has additionally been studied in the context of wireless systems to reliably transmit short packets of URLLC applications in [45]. The authors attempted to determine an optimal URLLC service design, concluding that it should be based on a non-asymptotic joint coding and queueing analysis that focuses on tail probabilities (i.e., quantiles) rather than averages. Their paper demonstrated how to perform such an analysis and described its novelty, which lies in the fact that it takes into account undetected bit errors. The occurrence of these errors may have a negative impact in time-critical communication systems.

In applied settings, the PAoI is often studied for systems with complex service disciplines and tandem queues, as for such cases its simpler structure allows distributions to be estimated rather than average values. In [35], the authors considered tandem queueing systems of the M/M/1–M/D/1 and M/M/1–M/M/1 types with FCFS service discipline. Such models find applications in many practical scenarios. For example, in edge-enabled IoT, wireless transmission precedes computational tasks and tandem queues can be utilized to capture these specifics. The authors concluded that M/M/1–M/D/1 is characterized by a smaller mean PAoI than M/M/1–M/M/1 for the same traffic arrival rate and mean service time.

Several authors have investigated the impact of channel impairments on the AoI and PAoI metrics. In [46], a two-way data exchange system consisting of a BS and a smart device was studied. The idea in this scenario is that the BS is constantly powered and the smart device is not. The BS simultaneously transmits information to the smart device over a block fading channel that receives data and stores it in a buffer. After collecting a certain amount of data, it sends an acknowledgment. The authors studied the timeliness and efficiency of the system in terms of the AoI and data rate. In [47], a scenario was investigated in which a source transmits data to several receivers. The AoI metric is utilized to measure the relevance of data from the perspective of the receiver. The authors assumed that the BS does not have a feedback channel with the receivers, and as such broadcasts the data regularly. They observed that when the transmission delay is random, the optimal update inter-generation interval needs to be larger than the expected transmission delay in order to successfully serve more users. Both theoretical analysis and simulation results showed that the average AoI performance under the proposed policy is close to the one when the system utilizes feedback from the receivers.

In practical studies, the AoI has been investigated for various scenarios that require timely updates. Specifically, the authors of [53] considered an unmanned aerial vehicle (UAV)-supported intelligent transportation systems for the delivery of time-critical traffic updates to vehicles. In [54], a URLLC scenario was addressed and a new system model was proposed for estimating the mean AoI for a URLLC wireless communication system that has a relay decoding and forwarding scheme over the Rayleigh block fading channels. In [61], the authors investigated the PAoI performance of URLLC services in 5G New Radio (NR) systems. They considered the queueing models in both discrete and continuous times by taking into account the specifics of OFDMA-based access. For the latter model, the mean PAoI time was derived. The simulation results showed that the PAoI behavior is mainly dictated by the loss performance required by the URLLC service. The calculated difference between the PAoI is minimal, often less than 10%; thus, when the loss guarantees are satisfied, the mean PAoI is insensitive to the choice of system parameters.

## 4. Open Challenges and Research Directions

In this section, we present an overview of relevant open challenges and research problems. We start with technology optimization for mMTC and URLLC services, and demonstrate that there are no the AoI/PAoI performance evaluation models developed to date that can be applied to such systems. Further, we proceed with theoretical challenges, highlighting the lack of general methods suitable for AoI/PAoI analysis.

### 4.1. Applications to Technology Optimization

The usage of the AoI/PAoI as metrics of interest is in its infancy, and practical technologies have yet to adopt AoI-based schedulers. However, as the number of state update applications grows there is increased interest in: (i) how conventional technologies perform with respect to this metric and (ii) what new functionalities need to be implemented to enable AoI/PAoI-sensitive application support in future technologies. This practically concerns the emerging applications of mMTC cellular IoT (CIoT) technologies, including LTE-M and NB-IoT systems such as energy grids [62,63], where the service requirements are much stricter compared to conventional environment monitoring. As of today there is ony a limited set of practical studies, leaving the question of designing a future sixth-generation (6G) radio interface supporting mMTC and URLLC AoI/PAoI-sensitive applications an open one.

#### 5G and 6G Cellular IoT Technologies

The requirements for conventional mMTC services designed for 5G systems are based on ITU-R M.2410, stating that the system should support one million devices per square kilometer with a minimum message intensity of one packet per two hours, a packet loss ratio (PLR) of 1%, and a maximum delay upper-bounded by 10 seconds, as detailed in the evaluation scenarios provided in ITU-R M.2412. Having these goals in mind, ITU-R has adopted a narrow-band Internet of Things (NB-IoT) interface for 5G mMTC service support [64]. This radio access technology (RAT) is aimed at a wide spectrum of 5G mMTC services.

The new emerging mMTC applications, such as those dictated by energy distribution systems, are characterized by principally different traffic patterns and QoS requirements. Operating under upper-layer management protocol suites such as supervisory control and data acquisition systems (SCADA; see Figure 8), these EDs are in the always-on “RRC-connected” state and their polling is centralized over regular time intervals. Such behavior induces the worst possible situation from a system performance point of view, having in mind batch arrival traffic patterns at the air interfaces that have been designed using purely stochastic arrivals. Even when access-barring techniques are utilized (which themselves negatively affect the latency performance of the system, as highlighted in [62]) or event-based activation is enforced, the presence of spatial correlation between sensor measurements may lead to batch traffic behavior at the air interface.

Modern LTE-M/NB-IoT RATs operate by utilizing a random access phase followed by a data transmission phase. Due to principally different requirements and dynamic traffic nature, similar to 5G NR, these phases need to be adaptively tuned to optimize performance metrics of interest induced by both random access and data transmission phases. The system operation can be represented as a two-stage tandem queueing system with multiple events (I—Idle, S—Success, C—Collision, E—Error); see Figure 9. The EDs successfully passing through the random access phase (RA) are scheduled for transmission in the data transmission (DT) phase of the frame. Here, one should not place constraints on the choice of preambles at the RA phase, and should consider the recent extension of the Shannon theorem for small data at the DT phase [65]. Note that during the RA phase errors might be caused by multiple EDs choosing the same preamble, or by the BS misinterpreting noise as preamble transmission. In the DT phase, errors are caused by propagation conditions or by multiple EDs being scheduled for transmission over the same resources. The illustrated system should be optimized with respect to the AoI and PAoI and explored to find the stability region providing the minimal values of these metrics of interest.

### 4.2. 5G and 6G URLLC Services

In industrial automation scenarios, 5G NR promises new applications, such as the joint operation of mobile robots, wireless time synchronization, positioning, augmented reality services for personnel, and telepresence-based maintenance operations [66]. The systems that control the moving elements of manufacturing equipment commonly generate low-rate traffic and require URLLC service, while video-guided machinery or mobile robots require eMBB service [67]. In these environments, 5G NR systems need to simultaneously support both services; notably, the the latter has extreme requirements in terms of packet drop probability, latency, and the AoI/PAoI.

One of the critical use cases for industrial automation scenarios is synchronization between entities. Time-Sensitive Communications (TSC) enhancements have to guarantee superior QoS as compared to wired technologies utilized for this purpose, such as process field network (PROFINET), EtherCAT, SErial Real-time COmmunication System (SERCOS), and Modbus [68]. According to the Third Generation Partnership Project Technical Specifications (3GPP TS) 22.104, the maximum end-to-end latency in such links should not exceed 1 ms and the mean time between failure should be ten years. Such strict performance guarantees can be achieved by utilizing either direct Device-to-Device (D2D) communications and/or NOMA. The rationale is that the scheduling unit in 5G NR, namely, the subframe, is exactly 1 ms in duration, preventing the utilization of ED-BS-ED communications.

When D2D connectivity is utilized, there should be a strict isolation of URLLC D2D traffic. Traffic coexistence support in 5G is technically enabled by network slicing functionality, which covers both wired and air interface domains [69]. As defined by 3GPP, a network slice is a logical network providing the prescribed set of networking capabilities [26]. According to 3GPP [69] and GSMA [70], there has to be both logical and physical isolation between the traffic of different slices. This isolation on the data plane can be ensured using mechanisms operating at different timescales, such as packet scheduling, connection admission control (CAC), network planning, etc. However, to date there have been no models proposed for capturing the AoI/PAoI performance of applications running over cellular technologies with OFDMA access. The reason for this is that such models should be based on queueing systems with polling mechanisms (see Figure 10), a part of queueing theory that has received little to no attention over the last decade. Such systems should account for the unstable nature of the air interface and implement a polling procedure similar to those utilized in 5G cellular systems, e.g., proportional fair or max–min [71,72].

Another approach to enable URLLC service in 5G and 6G systems is to utilize NOMA technology at the air interface to provide URLLC transmissions, with eMBB used in case of overloaded conditions. One of the inherent advantages of this approach is that URLLC data can be immediately scheduled without waiting until the end of the current NR frame for conventional scheduling. The major obstacle in modeling such systems comes from the assessment of the decoding probability. Similarly to the D2D case, no studies have attempted to evaluate the AoI/PAoI metrics in NOMA systems.

### 4.3. Theoretical Challenges

#### Realistic Traffic Patterns and Scheduling Strategies

Most of the studies addressing the AoI as a metric of interest have taken an analytical approach to system analysis. However, in this case the mathematical tractability of the systems affects the choice of the arrival and service processes as well as the service strategy. With respect to the arrival process, only systems with uncorrelated arrivals, i.e., the Poisson arrival process, can be easily analysed. However, the emergence of new applications within URLLC domains, and specifically in mMTC domains such as energy grids, calls for more complex arrival processes. For example, it is known that for low/high voltage (LV/HV) systems the EDs may generate highly correlated traffic patterns with batch arrivals, as demonstrated in [62,63]. Second, the service process may drastically deviate from the conventional exponential assumption, especially in wireless conditions, where retransmissions in the case of mMTC service may lead to more complex distributions. Contrarily, for URLLC service, where repetition-based coding is heavily utilized to meet the latency requirements, the service time is usually fixed, violating the common exponential assumption.

In addition to this, only the simplest service strategies, such as first in–first out (FIFO)/last in–first out (LIFO) and random service order, allow for elegant closed-form solutions. All of these have been used in the past. Specifically, as of now there is a clear lack of models that are directly applicable to wireless systems, including cellular technologies. Although there have already been studies designing AoI-based schedulers (see Section 3), the question of performance bounds in conventional cellular systems utilizing proportional fair and/or max-min schedulers remains an open one.

### 4.4. New Analysis Methods and Approaches

Despite the recent explosion in the number of studies related to the AoI, most of those remain based on conventional methods of analysis. Specifically, these studies heavily adopt the geometric interpretation of the AoI according to the Pollaczek–Khincine approach for the delay analysis in M/G/1 systems [73]; moreover, they utilize a direct arrival–epoch approach for the peak AoI and AoI distributions. However, such approaches are only applicable for a limited class of queueing systems having either arrival or service processes following memoryless exponential or geometric distributions. At the same time, the field of delay analysis in queueing theory is far richer, encompassing various approximations for queues having Markov-modulated inputs and/or service processes of phase type (PH). Owing to the versatility of the PH distributions, which are dense in the class of distributions over a non-negative axis [74], as well as that of Markovian arrival modes [75,76], extension to the case of matrix-geometric solutions seems feasible [77]. On top of this, various approximations and bounds developed for delay analysis can be extended to the case of the AoI [78].

We note here that the AoI, as a metric of interest, is more complicated to analyse mathematically than the delay. The major difficulty lies in the fact that this metric is often computed with respect to an arbitrarily chosen time instant in the AoI evolution. As a result, many authors instead consider the PAoI, representing those maximum values of the AoI associated with the event of packet arrival to the destination. Although such difficulties do not generally arise in simulation-based modeling, most of the studies carried out to date have concentrated on an analytical approach to PAoI characterization.

### 4.5. Queueing Networks

Many AoI-related studies nowadays either concentrate on a single queue between communicating entities or consider a single-queue approximation of the communications network. Both approaches are unrealistic for modern networks, where information is handled in many service elements while passing to the destination. However, the analysis of queueing networks for the AoI is limited to tandem queueing systems [79], with only a few notable exceptions [80]. Nonetheless, the analysis of Jackson-type queueing networks based on Burke’s theorem with various extensions is straightforward [81]. On top of this, approximate analysis of more comprehensive networks based on the decomposition approach can be utilized to address AoI as a metric of interest [82].

## 5. Conclusions

The AoI and PAoI are new performance measures that are useful for the characterization of services exchanging state update information. Depending on the application type, 5G cellular systems are expected to support two such types of service, namely, mMTC and URLLC. However, as both metrics were introduced only just over a decade ago, the research work in this area remains in its infancy, with studies addressing queueing-theoretic and practical systems in a rather ad hoc way.

In this paper, we provide: (i) a comprehensive introduction to these metrics, binding them to the specifics of the technical systems; (ii) a detailed review of the efforts made in this area to date from both theoretical and practical perspectives; and (iii) a discussion of identified open problems and research challenges for analysis of mMTC and URLLC systems using the AoI/PAoI as metrics of interest.

One of our critical observations is that no studies yet published are applicable to 5G OFDMA-based cellular systems, which are expected to become enablers for mMTC and URLLC services. To fill this gap, more comprehensive tandem queueing systems with polling and batch arrival and service processes need to be analysed. To this end, we demonstrate that the results of the AoI and PAoI immediately follow from systems that allow for delay analysis. Additionally, we demonstrate that most of the models proposed to date do not capture the specifics of mMTC and URLLC service delivery in 5G systems. Thus, appropriate models must be developed in order to enable truly AoI/PAoI-sensitive applications in the future evolution of cellular technologies.

## Figures and Tables

**Figure 1 sensors-23-08238-f001:**
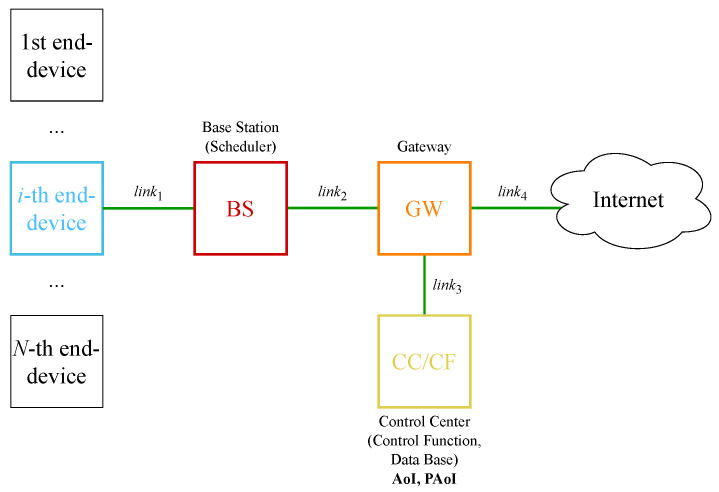
Illustration of the considered system model.

**Figure 2 sensors-23-08238-f002:**
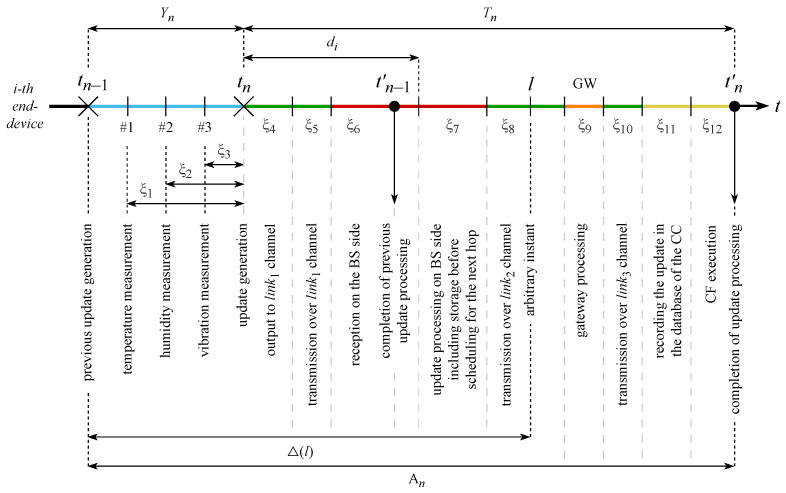
Detailed time diagram for transferring updates to the CC.

**Figure 3 sensors-23-08238-f003:**
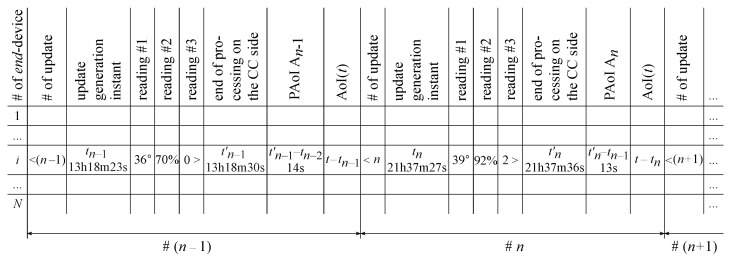
The information stored in the CC database.

**Figure 4 sensors-23-08238-f004:**
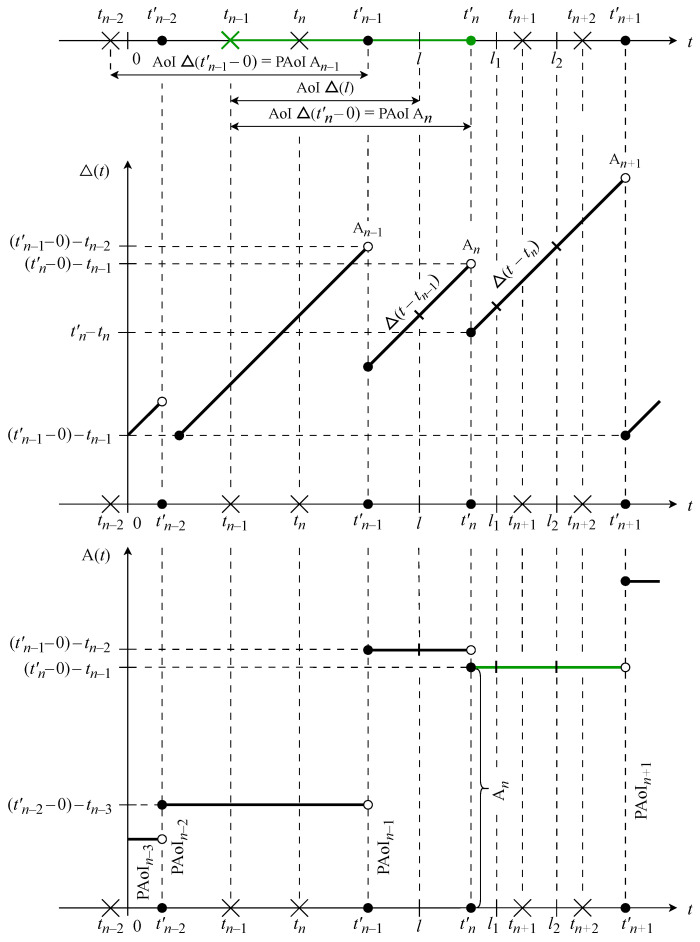
The AoI and the PAoI dynamics for a single ED.

**Figure 5 sensors-23-08238-f005:**
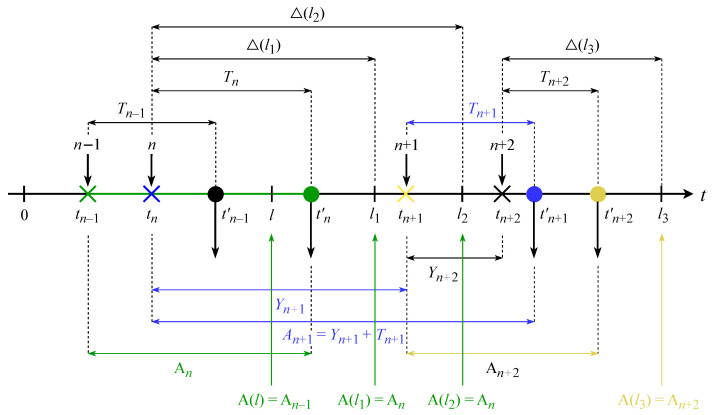
The AoI Δ(t) and the PAoI $A_{n}$, A(t) for a single ED.

**Figure 6 sensors-23-08238-f006:**
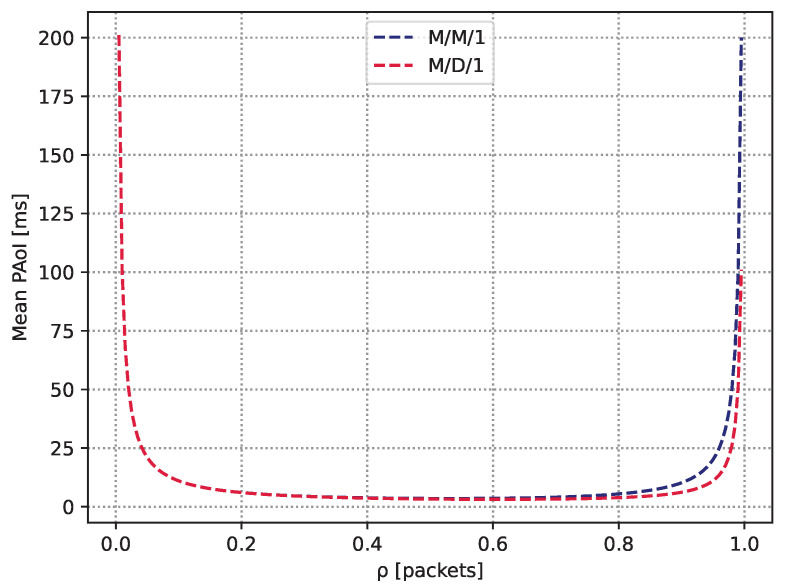
The mean PAoI in the considered M/M/1 and M/D/1 systems.

**Figure 7 sensors-23-08238-f007:**
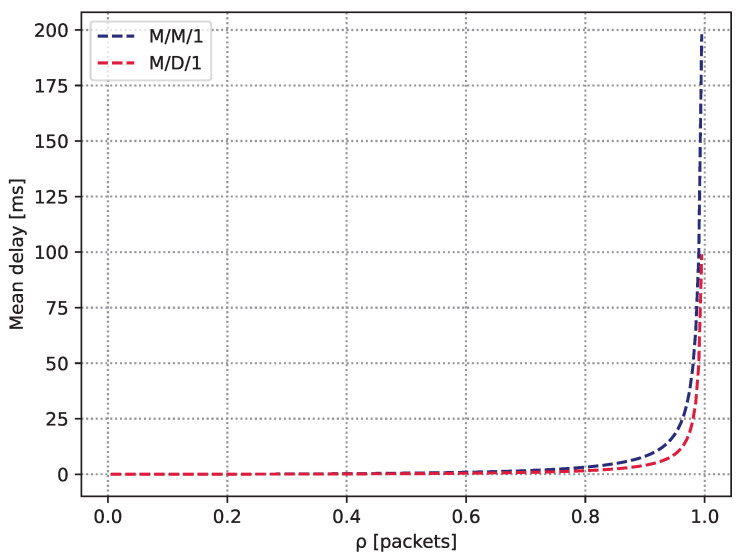
The mean full delay in the considered M/M/1 and M/D/1 systems.

**Figure 8 sensors-23-08238-f008:**
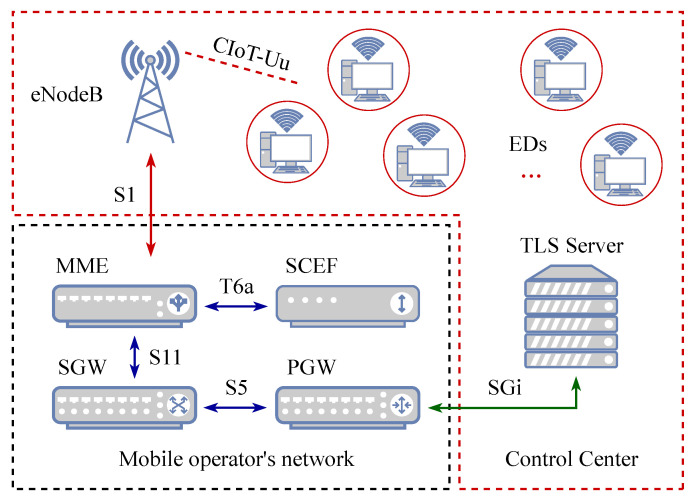
The overall framework for remote-control operation.

**Figure 9 sensors-23-08238-f009:**
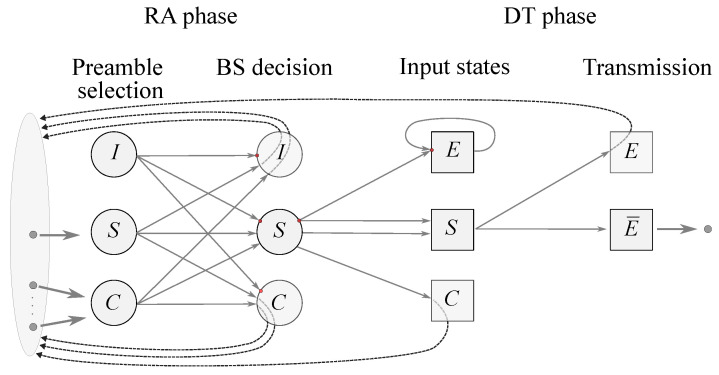
The possible events in 6G mMTC system: I—Idle; S—Success; C—Collision; E—Error.

**Figure 10 sensors-23-08238-f010:**
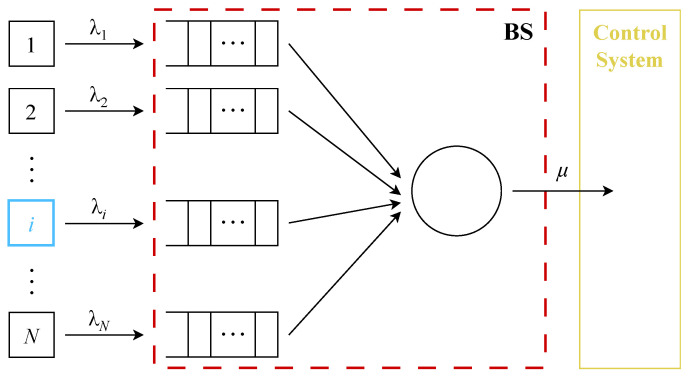
The polling service process over a wireless channel.

**Table 1 sensors-23-08238-t001:** Notation for a system with a single ED.

Notation	Meaning
*n*	update sequence number, n≥1
$t_{n}$	instant of generation of the *n*-th update on the ED side
$t_{n}^{\prime }$	instant of end of handling the *n*-th update on the CC side
$l,l_{1},l_{2},l_{3}$	arbitrary instants of access to the CC database
$\xi_{1},\xi_{2},\xi_{3}$	time intervals from the reading’s measurements to the start of the first bit of the corresponding update packet (the *n*-th update) transmission
$\xi_{4}$	serialization time at the ED
$\xi_{5}$	propagation delay over the wireless channel $link_{1}$
$\xi_{6}$	reception time of the *n*-th update at the BS
$\xi_{7}$	storage duration at the BS
$\xi_{8}$	transmission time over the high-speed $link_{2}$
$\xi_{9}$	handling time at the GW
$\xi_{10}$	transmission time over the high-speed $link_{3}$
$\xi_{11}$	handling a record time at the CC
$\xi_{12}$	CF execution time
$T_{n}$	*n*-th update delivery time, $T_{n}=\sum_{k=4}^{12}\xi_{k}$
$Y_{n}$	time before the *n*-th update generation since the previous update (n−1) generation
Δ(t)	the AoI at time instant *t*
$A_{n}$	the PAoI after handling the *n*-th update on CC side
A(t)	the current PAoI at time instant *t*

**Table 2 sensors-23-08238-t002:** Main characteristics of the reviewed studies.

Ref.	Queueing Model	Metr.	Analysis Type	Technical System
[17]	M/M/1, LCFS	AoI, AAoI	analytics	CSMA, 802.11 MAC
[28]	M/M/1, M/D/1, D/M/1, FCFS	AoI, AAoI	analytics	N/A
[38]	M/M/1, LCFS, with/without preemption service	AoI, AAoI	analytics	N/A
[39]	M/M/1, FCFS, LCFS	AoI	analytics	N/A
[40]	G/M/1, LCFS with/without preemption	AoI, AAoI, PAoI	simulation	relay networks
[41]	M/M/1/1, M/M/1/2, M/M/1/2∗	AoI, AAoI, PAoI	analytics, simulation	N/A
[42]	M/G/1/1	AoI, AAoI	simulation	IIR HARQ, FR HARQ
[43]	M/G/1/1 with preemption	AoI, AAoI, PAoI	simulation	–
[32]	M/M/1/1, M/M/1/2, M/M/1/2∗, FCFS	AoI, AAoI, PAoI	analytics, simulation	N/A
[44]	M/G/1	AoI, PAoI	simulation	heterogenous systems
[45]	Geo/G/1, $Geo^{\left[X\right]}/G/1$	AoI, PAoI	analytics	URLLC
[34]	Geo/Geo/1, discrete time, FCFS, LCFS	AoI, PAoI	simulation	N/A
[35]	M/M/1, M/D/1, FCFS	AoI, PAoI	simulation	edge computing
[46]	N/A	AoI, AAoI	simulation	wireless system
[47]	N/A	AoI	analytics, simulation	N/A
[48]	N/A	AoI, AAoI, PAoI	analytics, simulation	a single-hop wireless network
[49]	M/G/1/1	AoI, AAoI, PAoI	analytics	N/A
[50]	queueing network, M/M/1, LCFS	AoI	analytics	multi-hop networks
[51]	M/M/1	AoI	simulation	wireless broadcasting
[52]	Geo/G/1, FCFS, G/G/1, LCFS, discrete time	AoI, AAoI, PAoI	analytics	N/A
[53]	N/A	AoI	analytics	IoT
[54]	N/A	AoI, AAoI	analytics, simulation	IoT, URLLC
[55]	M/PH/1/1 with packet errors, M+M/M/1/1	AoI, AAoI, PAoI	analytics, simulation	N/A
[56]	M/G/1, a multi-source M/M/1, FCFS	AoI, AAoI	simulation	N/A
[57]	PH/PH/1/1, M/PH/1/2, M/PH/1/2∗ LCFS	AoI, PAoI	simulation	N/A
[58]	N/A	AoI, AAoI	analytics	N/A
[59]	Discrete time queueing systems	AoI, PAoI	simulation	N/A
[27]	GI/GI/1, FCFS and LCFS with preemption	AoI, PAoI	analytics	N/A
[60]	N/A	AoI, PAoI	analytics	N/A
[61]	$D^{X}/D^{X}/1/R$, $M/M^{X}/1/R$, FCFS	AoI, PAoI	simulation	URLLC

## Data Availability

Not applicable.

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
