# Peer review of "The Age of Information in Wireless Cellular Systems: Gaps, Open Problems, and Research Challenges"

_sensors, 2023, doi:10.3390/s23198238_

Round 1

Reviewer 1 Report

1. Fig 1 Should  should be corelated with the objectives mentions , as it is proposed modular diagram

2. It is unclear in what way the objectives are met- need to add details  with specific focus on mathematical and result oriented discussion

3. Need to Justify the claims in the abstract regarding AoI and PAoI-- missing 

4. Its one king of survey and no innovative idea   at least add the result of M/M/1 and M/D/1 what is effect 

 Need to rearrange sentences in meaningful way  by considering typo and grammatical mistakes 

Author Response

Dear Reviewer:

Thank you very much for sharing your expert opinions on our work. We really appreciate the time and effort taken in reviewing this submission. As a result, we are confident that our paper has benefited considerably from your constructive comments and suggestions, which were extremely helpful in improving the quality of this manuscript.

We have also made every effort to eliminate all of the indicated minor flaws and inconsistencies and do hope that as a result the quality of our manuscript has improved further. The focused point-to-point answers to the review comments follow, where the feedback from the Reviewer is given in regular font, while our comments are highlighted in bold font.

Reviewer 2 Report

My overall impression is that of the paper gives a comprehensive and thorough review of the concept of "Age of Information" in communication systems and networks. The paper gives to meticulously compile and summarize existing research and literature related to this topic. It covers a wide range of aspects, including mathematical modeling, queuing systems, optimization techniques, and applications in various communication scenarios.

For these reasons, this paper is a really promising paper and the content is likely of interest to your readers. However, there are lots of details missing and regarding strategies. I believe the changes outlined below will make this a really good paper.

The organization of the article is not well-structured or comprehensively described. Some parts of the document are disorganized or lacking in clarity, which makes it difficult for readers to follow the flow of information. While the document covers a wide range of related research and provides a list of references, the extent to which the article presents a significant contribution to the field might need further evaluation.

This could be solved if authors considered the follow:

The abstract is a mini paper; in this sense this must contain the main parts of the paper. The knowledge gap and the main result must be highlighted. I recommend reading this guideline from Nature about how to structure an effective abstract:

http://www.cbs.umn.edu/sites/default/files/public/downloads/Annotated_Nature_abstract.pdf

Introduction 

The GAP is a little buried, please clarify and highlight it. Please, authors have to think the Intro should answer some of the following questions: What is the problem to be solved? Are there any existing solutions? Which is the best? What is its main limitation? What do you hope to achieve? 

The English used in the article has issues that affect its readability. These issues include grammar, syntax, and vocabulary choices that can hinder the effective communication of ideas to the reader.

Author Response

(The authors gave the same response as above.)

Reviewer 3 Report

Among others, authors of this paper provide a study to identify the gaps between technical wireless systems and queueing models utilized for analysis of age of information (AoI) metric. The content of the article meets with the topics the SI planned to publish in the Sensors journal.

The basic topic of the article is interesting. However, its readability should be improved. Its major revision is mandatory.

Notes:

o   Introduction (and partly Section 3) – the main contributions of the article, related to the provided elaboration of the state-of-the-art (SoA), which is on general level, should be more highlighted.

o   Section 2 – typo in the English grammar – “We assume that that…”; next “The monitor located on the CC monitors the database…”.

o   Section 2 – the visibility of Fig. 3 is bad (it has bad sizes)

o   Section 2 – is it possible to complete this part with some simulation or measurement-based results? It should be nice the applying of AoI and PAoI on concrete examples.

o   Section 3 – Table 2 should be extended with the actual work of the authors.

Section 2 – typo in the English grammar – “We assume that that…”; next “The monitor located on the CC monitors the database…”.

Next, the text of the article contains some minor typos in the English grammar. Hence, I recommend for the authors to check the article carefully once again.

Author Response

(The authors gave the same response as above.)

Round 2

Reviewer 2 Report

Thanks for accepting the suggestions.

Author Response

Thank you very much for sharing your expert opinions on our work.

Reviewer 3 Report

The article has been improved. Many thanks for the explanation letter!

After checking the revied version of the article, I have the following notes:

-          Introduction – not all the used abbreviations are explained (e.g., LoRaWAN) – se the list of abbreviations.

-          Section 2.5 – sometimes, it is not exactly clear that what equations have been derived by the authors and what equations are used from other works.

-          Section 2.5 – Figs. 6 and 7 – the unites of parameters should be presented in closed brackets (“[]”).

-          Section 2.5 – it is not exactly clear that what system parameters were considered for the “simulation-based” analysis.

-          References – work [57] is not available online. Why?  

-     Introduction / References – in the Introduction, it should be also mentioned that the LoRaWAN networks can be also used not only in the sub-GHz, but also in the GHz (concretely 2.4 GHz) bands. For this reason, it might be possible to extend the references by the following works: "On the RSSI-based indoor localization employing LoRa in the 2.4 GHz ISM band” and “Ranging capabilities of LoRa 2.4 GHz”. Both works deal with possible utilization of the LoRa technology in the mentioned band (not only for a simple data transmission). I Hope that these works can be helpful and can increase the quality of the elaboration of the sate-of-the-art provided in this work. Otherwise, work with other articles. Thanks!

Author Response

(The authors gave the same response as above.)
